# Effects of Ionic Strength on Lateral Particle Migration in Shear-Thinning Xanthan Gum Solutions

**DOI:** 10.3390/mi10080535

**Published:** 2019-08-15

**Authors:** Mira Cho, Sun Ok Hong, Seung Hak Lee, Kyu Hyun, Ju Min Kim

**Affiliations:** 1Department of Energy Systems Research, Ajou University, Suwon 16499, Korea; 2School of Chemical and Biomolecular Engineering, Pusan National University, Busan 46241, Korea; 3Department of Chemical Engineering, Ajou University, Suwon 16499, Korea

**Keywords:** microfluidics, viscoelasticity, particle focusing, shear-thinning, xanthan gum, cell counting and sorting

## Abstract

Viscoelastic fluids, including particulate systems, are found in various biological and industrial systems including blood flow, food, cosmetics, and electronic materials. Particles suspended in viscoelastic fluids such as polymer solutions migrate laterally, forming spatially segregated streams in pressure-driven flow. Viscoelastic particle migration was recently applied to microfluidic technologies including particle counting and sorting and the micromechanical measurement of living cells. Understanding the effects on equilibrium particle positions of rheological properties of suspending viscoelastic fluid is essential for designing microfluidic applications. It has been considered that the shear-thinning behavior of viscoelastic fluid is a critical factor in determining the equilibrium particle positions. This work presents the lateral particle migration in two different xanthan gum-based viscoelastic fluids with similar shear-thinning viscosities and the linear viscoelastic properties. The flexibility and contour length of the xanthan gum molecules were tuned by varying the ionic strength of the solvent. Particles suspended in flexible and short xanthan gum solution, dissolved at high ionic strength, migrated toward the corners in a square channel, whereas particles in the rigid and long xanthan gum solutions in deionized water migrated toward the centerline. This work suggests that the structural properties of polymer molecules play significant roles in determining the equilibrium positions in shear-thinning fluids, despite similar bulk rheological properties. The current results are expected to be used in a wide range of applications such as cell counting and sorting.

## 1. Introduction

Microfluidics-based analytical tools have recently attracted much attention, as very small volumes of samples are typically required and the whole process from sample preparation to analysis can be potentially integrated onto a single chip [1]. One such promising microfluidic technology is the miniaturized flow cytometer for particle counting and sorting, which can be used for a wide range of point-of-care tests, such as the diagnosis of blood-related diseases [2]. The key fluidic part in the flow cytometer is the particle focusing unit, which is harnessed to focus suspended particles in a narrow fluid stream for one-by-one particle counting or sorting [3]. Various particle focusing methods have been proposed based on external force fields (active method) or on the nonlinear flow dynamics in Newtonian or non-Newtonian viscoelastic flow (passive method) [3,4,5,6,7]. Recently, passive methods based on the inertial and/or viscoelastic particle migration mechanism have been extensively studied, as these methods can be implemented in relatively simpler single layered channels, as compared to the complicated channel structures required for the active methods [2,4,5,6,7].

Viscoelasticity-driven particle focusing has recently been utilized in passive methods for a wide range of applications, such as micron-sized particle/cell sorting and DNA molecule and nanoparticle manipulation [6,7,8,9,10,11]. Viscoelastic particle focusing can be implemented in extremely simple and straight channels such as capillary tubes [8,10,12,13]. In addition, the optimal flow rate for particle focusing and the equilibrium particle positions can be tuned based on the rheological properties of the particle-suspending viscoelastic fluids [8,10,14]. On the other hand, it has been argued that the shear-thinning viscosity of the suspending fluid plays a critical role in determining the equilibrium particle locations: e.g., the channel centerline or wall [12,14,15,16,17]. Whether the equilibrium particle positions depend on the shear-thinning behavior has been an important issue since inceptive studies on lateral particle migration in viscoelastic fluids [16]. Numerical studies also suggested that the equilibrium particle positions are significantly affected by the shear-thinning viscosity of the particle-suspending medium [18]. More recently, it was observed that a portion of particles migrate toward the channel wall in the shear-thinning fluid in a cylindrical microtube, whilst this phenomenon was not observed in viscoelastic fluids with constant shear viscosity (Boger fluids) [12]. It was also revealed that the equilibrium particle positions in rectangular microchannels, which are relevant for most lab-on-a-chip applications, differ significantly depending on the shear-thinning fluid type, channel shape, and the presence of the inertial effect [12,14,15,19,20]. However, previous studies have been limited to the effects of the bulk rheological properties on lateral particle migration. There are no studies on the effect of the molecular structure of polymers on lateral particle migration.

In this work, we investigate lateral particle migration in a square microchannel (Figure 1a) in an aqueous solution of xanthan gum (XG) as a shear-thinning fluid. XG solution has been long used as a model shear-thinning fluid for various rheological studies and blood analogues [21,22,23,24,25,26,27,28]. The XG molecule is a polysaccharide with negative charge along its backbone and its native form is double-helical [29]. Single-molecule experiments showed that the ionic strength of the solvent significantly affects the structural properties of the XG molecule [30]. Both the persistence and contour lengths of the XG molecule decrease with increasing ionic strength of the dissolving solvent, which originates from screening of the intramolecular electrostatic repulsion interaction [30]. Therefore, it is expected that the rheological properties of XG solution can be tuned by varying the XG concentration and the ionic strength of the solvent. 

We prepared two XG aqueous solutions with very similar shear-thinning and linear/nonlinear viscoelastic properties that differed in terms of the structural properties of the XG molecules. The two solutions were prepared by modulating the XG concentration and the ionic strength of the solvent. We demonstrated that the equilibrium particle locations in the shear-thinning fluid are clearly affected by the ionic strength (or structural properties) of the XG molecules (Figure 1b).

## 2. Experimental

### 2.1. Microchannel and Materials

Lateral particle migration in a viscoelastic fluid was observed in a 5 cm long square microchannel (width (*w*) × height (*h*) = 50 μm × 50 μm), as schematically illustrated in Figure 1a.

The microchannel was fabricated with poly(dimethyl siloxane) (PDMS) by utilizing the conventional soft lithography technique [31] (refer to our previous work [8] for specific conditions).

Herein, the following two shear-thinning fluids were prepared by dissolving XG (Sigma-Aldrich, St. Louis, MO, USA) of two different concentrations in deionized (DI) water or 1× phosphate buffered saline (PBS) solution (ionic strength = 0.163 M): 0.05 wt % XG solution in DI water (XGDI) and 0.1 wt % XG solution in PBS solution (XGPBS). Shear viscosity and small-amplitude oscillatory shear (SAOS) tests were performed at 20 °C using a stress-controlled rheometer (ARES-G2, TA Instruments, New Castle, DE, USA) equipped with cone-and-plate geometry (1°; diameter: 60 mm). Large-amplitude oscillatory shear (LAOS) tests were also performed at 20 °C using a strain-controlled rheometer (ATES-G2, TA Instruments) equipped with cup and bob accessories (cup inner diameter: 30 mm; bob outer diameter: 27.7 mm); the frequency was set to 1 rad·s^−1^. The raw stress data were decomposed into elastic and viscous shear stress as described elsewhere [32]. The measured shear viscosity of the XG solution ($\mu_{\mathrm{XG}}$) was modeled by applying the Carreau model [33] as follows:(1)$$\mu_{\mathrm{XG}}=\mu_{\inf}+\left(\mu_{0}-\mu_{\inf}\right){\left[1+{\left(\lambda \dot{\gamma }\right)}^{2}\right]}^{\frac{n-1}{2}}$$
where $\mu_{0}$ is the zero-shear viscosity, $\mu_{\inf}$ is the infinite-shear viscosity, λ is the relaxation time, $\dot{\gamma }$ is the shear-rate, and *n* is the power-law index. For reference, a 6.4 wt % poly(vinyl pyrrolidone) (PVP) (M.W. = 360,000 g·mol^−1^, Sigma-Aldrich) aqueous solution was also prepared and used as the constant shear viscosity (μ) viscoelastic fluid. The concentration of PVP (6.4 wt %) was adjusted to keep the viscosity of the PVP solution close to the zero-shear viscosities ($\mu_{0}$) of the XG solutions. The relaxation time of the PVP solution was obtained by fitting the storage (G’) and loss (G’’) moduli, which were measured with the SAOS test by applying the Maxwell model [33]. The rheological properties of the two XG (XGDI, XGPBS) and PVP solutions are presented in Table 1.

### 2.2. Microfluidics, Imaging, and Flow Characterization

For the lateral particle migration experiments, 0.04 vol % polystyrene (PS) microspheres (diameter = 10 μm; Polyscience) were added to the polymer solutions. The viscoelastic fluids including the PS beads were flowed through the inlet (Figure 1a) and the volumetric flow rate (Q) was controlled with a syringe pump (11 Plus, Harvard Apparatus Inc., Holliston, MA, USA). The microchannel was pre-treated prior to the particle migration experiments by flowing 0.1 wt % nonionic surfactant (Tween 20, Sigma-Aldrich) aqueous solution at Q = 10 μL·h^−1^ for more than 10 min to minimize the particle (PS bead) adhesion on the channel walls. Images were captured 4 cm downstream from the inlet with a high-speed camera (FASTCAM MC2, Photron, Tokyo, Japan) under a 10× objective installed on an upright microscope (BX-60, Olympus, Tokyo, Japan). The acquired images were processed with ImageJ software (National Institutes of Health, Bethesda, MD, USA) and the locations of the PS beads were determined as previously described [8]. 

The pressure drop (Δ*P*) between the inlet and outlet in the microchannel was measured as a function of the flow rate. Two commercial pressure sensors specialized for microfluidics (uPS0250 and uPS1800, LabSmith, Inc., Livermore, CA, USA) were used to measure the pressure drop over a wide range (0 ≤ Δ*P* ≤ 602 kPa). The pressure sensor was installed in a tubing between the microchannel inlet and the syringe, close to the channel inlet (distance ≈ 1 cm). Therefore, it is assumed that the pressure drop between the pressure sensor location and the microchannel inlet is negligibly small because the tubing inner diameter (760 μm) is much larger as compared to the microchannel height (50 μm). The pressure drop at each flow rate was obtained by averaging the 3000 pressure values measured at 0.1 s intervals after the flow reached steady-state. In a rectangular channel, the pressure drop for a Newtonian fluid is predicted to be $\Delta P=\frac{\alpha \mu QL_{c}}{wh^{3}},$ where α is defined as $12\times {\left[1-\frac{192h}{\pi^{5}w}\tanh\left(\frac{\pi w}{2h}\right)\right]}^{-1}$ and $L_{c}$ is the channel length [34,35]; the calculated values were compared with the experimental data measured using PVP solution.

The viscoelastic flow was characterized with two dimensionless groups: Reynolds (*Re*) and Weissenberg (*Wi*) numbers. The Reynolds number denotes the relative ratio of inertial to viscous forces, defined as Re=ρ〈u〉L/μ (*ρ*: Fluid density; $\langle u\rangle \left(\equiv Q/h^{2}\right)$: Average velocity; L(≡h): Characteristic length scale). The zero-shear viscosity, $\mu_{0},$ was adopted as μ for the shear-thinning XG solutions. The Weissenberg number, defined as $Wi=\lambda {\dot{\gamma }}_{c}$, represents the ratio of elastic to viscous properties where ${\dot{\gamma }}_{c}\;\left(\equiv 2Q/h^{3}\right)$ is the characteristic shear-rate [33]. The non-dimensionless groups calculated based on the channel geometry and the flow rates considered in this work are presented in Table 2.

## 3. Results and Discussion

Herein, 10 μm PS beads flowed through a square channel, as shown in Figure 1a. The lateral particle migration was investigated in three different viscoelastic media: 0.05 wt % XG solution in DI water (XGDI); 0.1 wt % XG solution in PBS solution (XGPBS); and 6.4 wt % PVP aqueous solution (PVP). 

The PS beads at the inlet were initially randomly distributed in the channel cross-section and the particles migrated toward different equilibrium positions according to the suspending media, as shown in Figure 1b. The shear viscosities depending on the shear rate are presented for these three fluids in Figure 2a and the linear viscoelastic properties are shown in Figure 2b. 

The rheological properties of these viscoelastic fluids are summarized in Table 1 (refer to the experimental section for the specific details of the viscosity modeling and the characterization methods). As shown in Figure 2a, the two XG solutions clearly exhibit shear-thinning behavior, whilst the PVP solution has nearly constant shear viscosity. In addition, the shear-thinning behavior of the two XG solutions was quite similar, as shown in Figure 2a and Table 1. In particular, the power-law index (0.56) of XGDI is quite close to that (0.58) of XGPBS (Table 1) and the linear viscoelastic properties (G’ and G”) of both fluids were also very similar, as shown in Figure 2b. 

The three-dimensional velocity fields in the XGDI, XGPBS, and PVP solutions were predicted at Q= 1 mL·h^−1^ with a commercial finite element method (FEM) simulation software (COMSOL Multiphysics^®^, COMSOL, Inc., Burlington, MA, USA) based on the rheological properties presented in Table 1. It was demonstrated that the velocity profiles in the shear-thinning fluids (XGDI and XGPBS; Figure 3a,b,d,e) are blunter than the quadratic profile of the viscoelastic fluid with constant shear viscosity (PVP; Figure 3c,f). In contrast, the velocity profiles for the two XG solutions (XGDI and XGPBS; Figure 3a,b,d,e) were predicted to be very similar. Nevertheless, the particle distributions in the two XG solutions were strikingly different, as shown in Figure 4. The PS beads in XGPBS migrated toward the channel corners, while the beads were focused along the channel centerline in XGDI. For comparison, the particle distribution in the PVP solution is presented in Appendix A and five equilibrium positions were observed along the channel centerline and four corners, consistent with the previous studies in Boger fluids [8,15,36]. 

Lateral particle migration was also previously studied in XG solutions in DI water, and it was demonstrated that the equilibrium particle positions varied according to the XG concentration, channel shape, and flow rate [19]. However, comparison of the previous results with the current work is not straightforward as the rheological properties of the XG solutions were not quantified in the previous study. Although reference was made to the shear viscosity data previously measured with the XG solutions in tap water [37], the rheological data for XG solution may vary significantly according to the ionic strength of the solvent, as discussed later.

There have been many studies on how the rheological properties of the suspending medium influence the equilibrium particle positions [6,8,12,13,14,16,18,19,20,30]. However, the specific conditions that determine the equilibrium particle positions in the shear-thinning fluid are not yet clear. For instance, it was reported that most particles were aligned along the centerline of a microtube, but some particles moved along the channel wall, while all the particles were aligned along the channel centerline in a Boger fluid [12]. On the other hand, it was reported that the particles in the viscoelastic fluid were focused in the middle-plane of a micro-slit channel regardless of whether the shear viscosity was constant or shear-thinning [13]. Therefore, the previous works demonstrate that the equilibrium particle position in shear-thinning fluids is a very complicated phenomenon related to both the rheological properties and the channel geometry [12,13,19].

The current experimental data show, at least, that the shear-thinning behavior is not sufficient to explain the determination of the equilibrium particle positions because both XGDI and XGPBS have very similar shear-thinning behavior with completely different equilibrium particle positions. Therefore, our experimental observation of the different equilibrium particle positions in the two XG solutions cannot be explained based on the existing theories or numerical simulation results, as most existing mathematical models of viscoelastic particle migration are based on constitutive equations for predicting the bulk rheological properties [12,13,17,38]. On the other hand, it was predicted that lateral particle migration in a viscoelastic fluid is driven by the imbalanced normal stress differences ($N_{1}$ and $N_{2}$) in the pressure-driven flow [39], where $N_{1}$ and $N_{2}$ denote the first and second normal stress differences, respectively. The magnitude of $N_{2}$ is usually assumed to be much smaller than that of $N_{1}$ in polymer solutions [13]. We note that it was not possible to directly measure the $N_{1}$ values of the XG solutions with the stress-controlled rotational rheometer. Instead, we investigated the nonlinear rheological properties with the large-amplitude oscillatory shear (LAOS) test (Figure 5).

LAOS tests are attractive rheological methods to distinguish complex fluids [40], which was also applied to XG solution [27]. At large strain amplitude, the shear stress curve distorted. For better analyzing distorted stress, the shear stress is decomposed into the elastic and viscous shear stresses [26], but there was also no significant difference in both elastic and viscous shear stresses between the two samples (Figure 5). We found that the elastic stresses of XGDI and XGPBS are slightly different from each other as compared to the viscous shear stress. We speculated that the elastic stress may more strongly affect the lateral motion of particles in the shear-thinning fluids. As mentioned before, however, it is difficult to conclude easily because the difference is not large.

It was previously reported that the measured persistence length of renatured XG molecules is ~400 nm in DI water, but decreases to ~150 nm when the ionic strength increases to 0.1−0.5 M [30]. The contour length also decreases from ~2 μm in DI water to ~600−700 nm as the ionic strength increases to 0.1−0.5 M [30]. In addition, it was observed that the single-helix and double-helix forms of the renatured XG molecules were mixed in DI water, whilst the double-helix form was only observed when electrolyte was added to the DI water [30]. Therefore, the structural properties of the XG molecules in the two solutions are expected to be significantly different, even though the two solutions have very similar bulk rheological properties, as shown in Figure 2 and Figure 5. The current experimental results suggest that the differences in the molecular conformation may lead to the different equilibrium particle positions.

Deformable materials such as flexible polymers and red blood cells (RBCs) migrate toward the channel centerline in the Poiseuille flow of a Newtonian fluid because of the hydrodynamic interactions between the materials and the channel walls [41,42,43,44]. In microcirculation flows, RBCs migrate toward the centerline while white blood cells (WBCs) are separated from RBCs and move along the channel walls in a process termed WBC margination [44]. WBC margination originates from the difference in the lift force between the two blood cell species: the deformable and non-spherical RBCs migrate toward the channel centerline because of cell wall hydrodynamics, while the lateral migration motion of stiff and spherical WBCs is not significant compared to that of the RBCs [45]. The very different equilibrium particle positions observed in the two XG solutions can be explained by the difference in the lift forces due to the differences in the flexibility of the XG molecules. We postulate that the flexible XG molecules in XGPBS tend to migrate toward the channel centerline, which results in margination of the rigid spherical particles as observed in Figure 4. Particle migration toward the channel corners was also observed in a shear-thinning fluid [46]. In contrast, the centerline focusing observed in DI water can be attributed to the synergistic combination of migration driven by the normal stress difference and secondary flow effects, where the latter is generated by the second normal stress difference [17] and the lateral migration of the stiff XG molecules is assumed to be insignificant. Centerline focusing in shear-thinning fluids was previously predicted and also observed experimentally [14,17]. Focusing of particles at the equilibrium positions along the channel centerline or the corners was also found in different shear-thinning fluids and under different flow conditions [14,46]. However, the present results demonstrate that particles migrate toward the centerline or corners even in the shear-thinning fluids with apparently similar rheological properties and under similar flow conditions. 

The present results demonstrate that the equilibrium particle positions in the shear-thinning fluid can be tuned by varying the ionic strength of the suspending medium. Tuning of the equilibrium particle positions in shear-thinning fluids is expected to be useful in a wide range of applications. As shown in Figure 4a,c (also refer to Appendix A), centerline focusing in DI water was achieved up to a high flow rate of ~2 mL·h^−1^, where this phenomenon can be exploited in high-throughput particle counting and separation [10]. On the other hand, the corner-entrainment of the rigid particles observed in the PBS solution (Figure 4c,d and Appendix A) can be applied to “deformability-selective cell entrainment” [36]. “Deformability-selective cell entrainment” was originally demonstrated in a viscoelastic fluid with constant shear viscosity (PVP solution) [36]. However, the number of entrained cells moving along the corners of the square channel was quite limited as most cells moved along the channel centerline in the PVP solution. In contrast, we expect that cell recovery can be significantly enhanced by employing XGPBS solution as a viscoelastic medium, as most particles are entrained along the channel corners in XGPBS. Particle focusing along the equilibrium positions was notably deteriorated in both XGDI and XGPBS when Q > ~2 mL·h^−1^, as shown in Figure 4 and Appendix A. The weakening of the particle focusing above Q = ~2 mL·h^−1^ can be attributed to increasing relevance of the inertial effect with increasing flow rate, e.g., Re ≈ 1 at Q = 8 mL·h^−1^ (Table 2), which was also observed in a viscoelastic fluid with constant shear viscosity [8]. We mention that the current study can be directly applied to the non-biological samples such as polymer particles and the PBS-based solution can be also used for the “deformability-selective cell entrainment” without the alteration of the solvent. However, the addition of non-ionic sucrose to the DI water-based solution can be considered to satisfy the osmotic balance between cells and the solvent, if the DI-based xanthan solution is applied to cell counting or sorting. In this work, we considered the particle dynamics only in a square microchannel, but it would be interesting to study the effects of the geometrical shape (ratio of channel width to channel) on the equilibrium particle positions. Finally, the required pressure drops for device operation were significantly lower in the shear-thinning fluids (XGDI and XGPBS; the two data sets almost overlapped in the figure) as compared to those in PVP solution (Appendix A), which demonstrates that high-flow-rate operation is possible with the shear-thinning fluids. 

## 4. Conclusions

The equilibrium particle positions in shear-thinning XG solutions can be changed considerably by varying the ionic strength of the solvent, despite similarities in bulk rheological properties such as shear viscosity and linear viscoelastic properties, as measured with conventional rotational rheometers. It is proposed that the interactions between XG molecules and rigid spheres may play a significant role in determining the equilibrium particle positions in a manner analogous to WBC margination. The current experimental results at least corroborate the idea that shear-thinning behaviors measured with conventional rheometers are not the sole determinant of equilibrium particle positions. It is anticipated that tuning equilibrium particle positions by controlling the ionic strength of XG solutions will be exploited in various practical applications such as particle counting and sorting and “deformability-selective” cell sorting [36]. 

## Figures and Tables

**Figure 1 micromachines-10-00535-f001:**
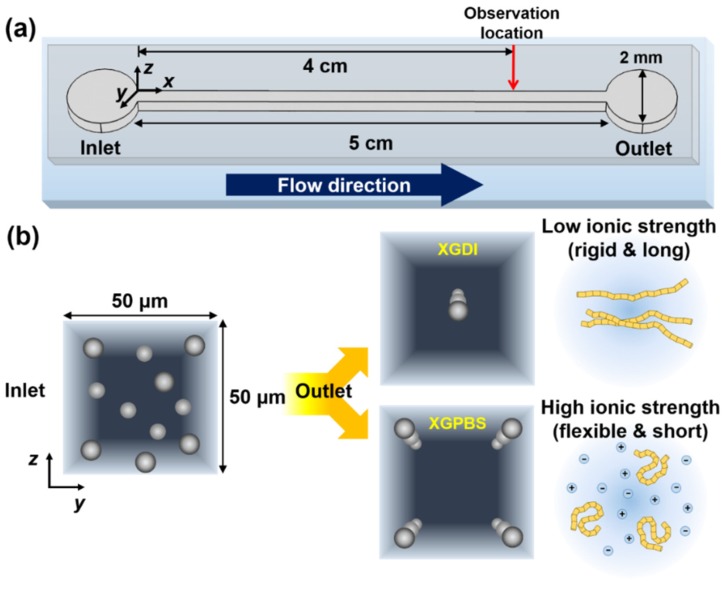
(**a**) Schematic of 5 cm long straight, square microchannel (width × height = 50 μm × 50 μm). The viscoelastic fluid including the polystyrene (PS) beads was injected through the inlet and the distribution of the PS beads (diameter = 10 μm) was observed 4 cm downstream from the inlet. (**b**) Schematic diagrams of random particle distributions at the inlet, and the equilibrium particle positions 4 cm downstream from the inlet in the xanthan gum (XG) solutions in deionized (DI) water (XGDI) or phosphate buffered saline (PBS) solution (XGPBS). The XG molecules adopt different molecular structures according to the solvent ionic strength: rigid and long in DI water; flexible and short in PBS solution [26].

**Figure 2 micromachines-10-00535-f002:**
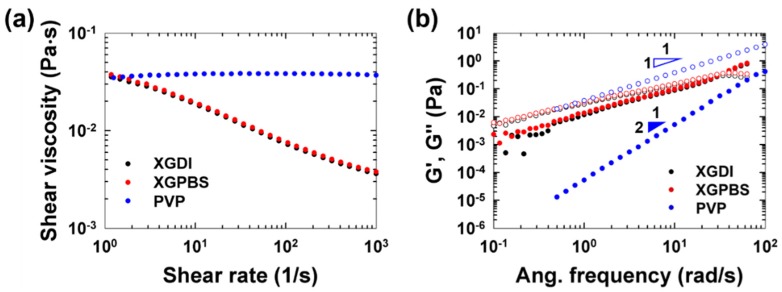
Rheological properties measured with a rotational rheometer (1° cone-and-plate geometry with 60 mm diameter) at 20 °C. (**a**) Steady shear viscosities of the xanthan gum (XG) solutions (XGDI and XGPBS) and poly(vinyl pyrrolidone) (PVP) solution. (**b**) Small-amplitude oscillatory shear test data (linear viscoelastic properties) of the XG solutions (in deionized water (XGDI) or PBS solution (XGPBS)) and PVP solution: G’ (storage modulus) and G” (loss modulus). The slopes in (**b**) demonstrate that the linear viscoelastic properties of the PVP solution follow terminal behaviors.

**Figure 3 micromachines-10-00535-f003:**
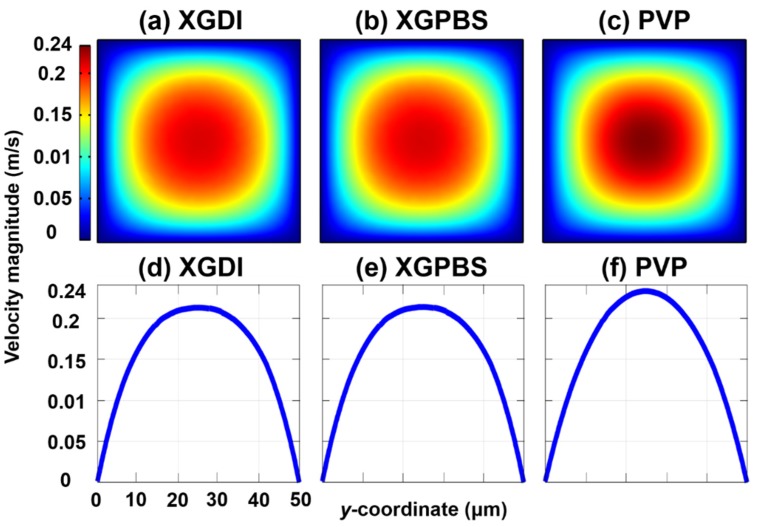
Numerical simulations for viscoelastic fluids i.e., shear-thinning xanthan gum (XG) solutions (XGDI and XGPBS) and viscoelastic fluid with constant shear viscosity (PVP) using a commercial finite element method software (COMSOL Multiphysics^®^) at Q = 1 mL·h^−1^, where the Carreau and Newtonian constitutive equations were employed for the XG solutions and PVP solution, respectively. (**a**–**c**) The upper contour diagrams denote the velocity magnitude distributions in the cross-section of XGDI, XGPBS, and PVP, respectively. (**d**–**f**) The lower graphs denote the stream-wise velocity magnitude distributions of XGDI, XGPBS, and PVP at the channel mid-plane, respectively.

**Figure 4 micromachines-10-00535-f004:**
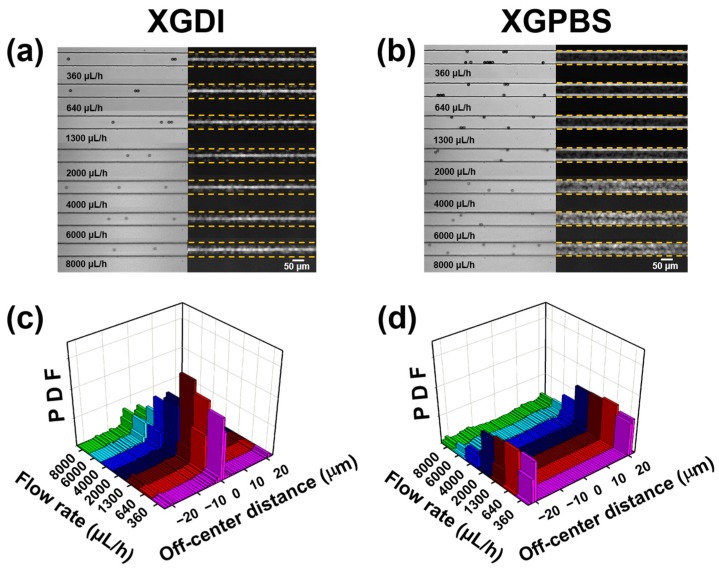
(**a**) Particle distributions in xanthan gum (XG) solution in deionized water (XGDI): snapshot images (left column) and standard deviation images (right column). (**b**) Particle distributions in XG solution in phosphate buffered saline (PBS) solution (XGPBS): snapshot images (left column) and standard deviation images (right column). (**c**) Particle distribution function (PDF) in XGDI. (**d**) PDF in XGPBS. The images were captured 4 cm downstream from the channel inlet. The standard deviation images were obtained by stacking 2000 successive time-lapse images with the standard deviation option using ImageJ software. The particle images were slightly blurred at high flow rates (Q≥ 4000 μL/h), but there was no significant problem in determining the particle location.

**Figure 5 micromachines-10-00535-f005:**
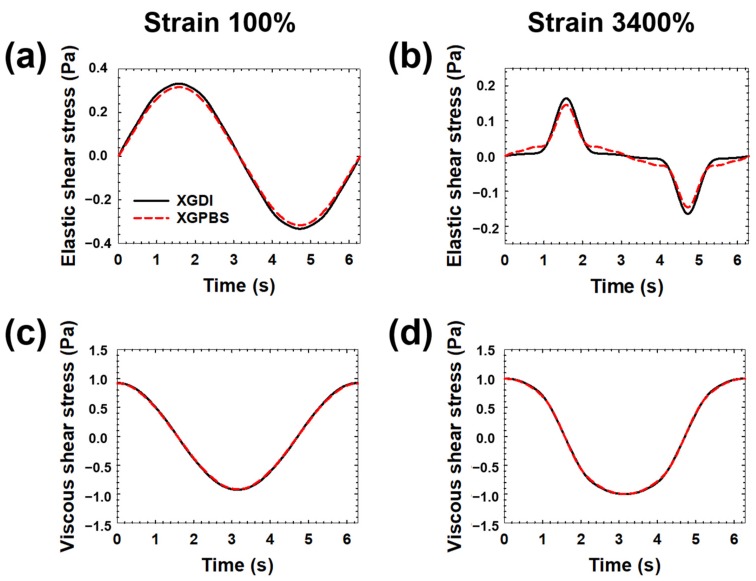
Large-amplitude oscillatory shear test data for the xanthan gum solutions (XGDI and XGPBS) at the fixed frequency of 1 rad/s and 20 °C. The stress measured at strain = 100% (left column) and 3400% (right column) was decomposed into the elastic (**a**,**b**) and viscous (**c**,**d**) strain as previously described [32].

**Table 1 micromachines-10-00535-t001:** Rheological properties of viscoelastic solutions.

Property		XGDI	XGPBS	PVP
Zero-shear viscosity (cP)	*μ* _0_	38.9	45.6	38.1
Infinite-shear viscosity (cP)	*μ* _inf_	1.3	1.7	
Power-law index	*n*	0.56	0.58	
Relaxation time (ms)	*λ*	34.2	29.3	1.4

Xanthan gum solutions (XGDI and XGPBS) were modeled with Carreau model, and PVP solution was modeled with single mode Maxwell model.

**Table 2 micromachines-10-00535-t002:** Dimensionless groups of Reynolds (Re) and Weissenberg (Wi) numbers calculated based on the channel geometry and the flow rates considered in this work.

Q (mL·h^−1^)	XGDI		XGPBS		PVP	
	*Re*	*Wi*	*Re*	*Wi*	*Re*	*Wi*
0.2	-		-		0.0292	1.2
0.36	0.0514	54.6	0.0439	46.9	0.0525	2.2
0.64	0.0914	97.1	0.0780	83.3	0.0933	3.9
1.3	0.1857	197.3	0.1584	169.2	-	
2	0.2856	303.6	0.2437	260.4		
4	0.5713	607.1	0.4873	520.7		
6	0.8569	910.7	0.7310	781.1		
8	1.1425	1214.2	0.9747	1041.4		

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
