# Peer review of "Effects of Ionic Strength on Lateral Particle Migration in Shear-Thinning Xanthan Gum Solutions"

_micromachines, 2019, doi:10.3390/mi10080535_

Round 1

Reviewer 1 Report

This manuscript (MS) shows the lateral particle migration in square microchannel of two different xanthan gum-based viscoelastic fluids to be used in microfluidic devices for cell counting and sorting. This MS is generally clearly presented and is under the scope of the Journal. However, several issues need to be addressed before the acceptance of this MS.

The following points need to be addressed:

Major

The authors mentioned that an important application of the proposed fluids is related to blood cells counting and sorting. Hence should be relevant to include in the literature review some blood analogues studies performed with xanthan gum-based viscoelastic fluids. Examples of some relevant works are the research performed by Sousa et al., Biomicrofluidics 5(1), 014108, 2011; Campo-Deano et al., Biomicrofluidics 7(3), 034102, 2013; Calejo, Micromachines 7(1), 4, 2016 and Pinho et al., Biomicrofluidics, 11, 054105, 2017. Improvements should be made on this point.

The authors mentioned that “It was also revealed that the equilibrium particle positions in rectangular microchannels, which are relevant for most lab-on-a-chip applications, differ significantly depending on the shear-thinning fluid type, channel shape, and the presence of the inertial effect”. However, this study was only performed for a square microchannels. In this way, it is strongly recommended to include a new section about challenges and future directions and should be included what will happen in rectangular microchannels.

What is the size of the particles used in this study? What will happen if you have a fluid with different size of particles and what about different shapes. If you have red blood cells (RBCs) instead of rigid particles what will happen? Can you suspend RBCs into the proposed fluid without the cells suffer any molecular damage? It will be nice to include these topics within the discussion or at the new section about challenges and future directions. The study performed by Pinho et al., Biomicrofluidics, 11, 054105, 2017 and the review performed by Bento et al., Micromachines 9(4), 2018, may help to answer to some of these questions.

Minor

Please include in Figure 1 the size of particles.

In Figure 4 a) and b) it is difficult to see the particles. Please include a detail in both figures.

Author Response

Reviewer #1: The authors mentioned that an important application of the proposed fluids is related to blood cells counting and sorting. Hence should be relevant to include in the literature review some blood analogues studies performed with xanthan gum-based viscoelastic fluids. Examples of some relevant works are the research performed by Sousa et al., Biomicrofluidics 5(1), 014108, 2011; Campo-Deano et al., Biomicrofluidics 7(3), 034102, 2013; Calejo, Micromachines 7(1), 4, 2016 and Pinho et al., Biomicrofluidics, 11, 054105, 2017. Improvements should be made on this point.

Thank you for informing us of the blood analogues research and we added the references in the revised manuscript (page 2), as suggested.

The authors mentioned that “It was also revealed that the equilibrium particle positions in rectangular microchannels, which are relevant for most lab-on-a-chip applications, differ significantly depending on the shear-thinning fluid type, channel shape, and the presence of the inertial effect”. However, this study was only performed for a square microchannels. In this way, it is strongly recommended to include a new section about challenges and future directions and should be included what will happen in rectangular microchannels.

Thank you for your comments. In this work, we studied the particle dynamics in a non-circular channel in shear-thinning xanthan solutions. As commented by the reviewer, we only considered the square channel but the squared-shape has been extensively employed in microfluidics community, which enables the direct comparison of this work with the previous works (e.g., refs. 8 and 35). Nevertheless, it would be interesting to study the geometrical effects on the lateral particle migration in the shear-thinning fluids used in this work, which would be helpful to the channel design for the practical applications. We added a note on this issue in page 10.

What is the size of the particles used in this study?

The particle size was 10 micron in diameter.

What will happen if you have a fluid with different size of particles and what about different shapes.

The lateral particle migration speed is proportional to  in viscoelastic fluid (ref. 35). Therefore, we expect that the particle focusing efficiency in the shear-thinning fluids also depends upon the square of the particle size. In the previous work (Kim et al., Anal. Chem., 89, p8662, 2017), the non-spherical particles were also focused along the channel centerline in the viscoelastic fluid with constant shear viscosity. We expect that the non-spherical particles were similarly focused along the equilibrium positions in the shear-thinning fluids, however, which demand further studies to find an optimal flow rate for the particle focusing.

If you have red blood cells (RBCs) instead of rigid particles what will happen?

It was previously demonstrated that the RBCs were focused along the channel centerline by the combination of the normal stress difference-induced and the deformability-induced migration effects in viscoelastic fluid (ref. 35). We expect that the RBCs can be also focused along the equilibrium positions in shear-thinning fluid, if the channel dimension is appropriately optimized and the solvent recipe is changed to satisfy the osmotic balance as noted in page 9. It would be interesting to study the effects of the RBC deformability on the particle focusing in the shear-thinning fluids.

Can you suspend RBCs into the proposed fluid without the cells suffer any molecular damage? It will be nice to include these topics within the discussion or at the new section about challenges and future directions. The study performed by Pinho et al., Biomicrofluidics, 11, 054105, 2017 and the review performed by Bento et al., Micromachines 9(4), 2018, may help to answer to some of these questions.

Thank you for your comments and information. As previously demonstrated (ref. 35), the RBCs, suspended in 1 PBS solution, will not be damaged since its ionic strength is isotonic and the addition of non-ionic sucrose to the DI-based solution can be considered to satisfy the osmotic balance between cells and solvent, if the DI water-based solution is applied to the cell-related experiments. We added the corresponding sentences at the end of page 9.

Please include in Figure 1 the size of particles.

We added the particle size in the figure caption of Fig. 1.

In Figure 4 a) and b) it is difficult to see the particles. Please include a detail in both figures.

As commented by the reviewer, the particle images at the high flow rates are slightly blurred, but there was no problem in the determination of the particle location. We added a note on this issue in the figure caption of Fig. 4.

Reviewer 2 Report

In this work the authors investigate the lateral particle migration in a square microchannel an aqueous solution of xanthan gum, which is a shear-thinning fluid. They find that the contour lengths of the XG molecule decrease with increasing ionic strength of the PBS saline, which has the consequence that XG molecules are clearly affected by the ionic strength of the liquid. The authors perform a particle flow experiments, using PS beads, Comsol simulations, and extensive rheological  characterization of the liquids.

The paper is well written, and the results are clearly presented in well composed figures. The conclusions seem to be appropriately supported by experiments and modelling.

In conclusion, I recommend publication of the ms in its present form.

Author Response

Reviewer #2In this work the authors investigate the lateral particle migration in a square microchannel an aqueous solution of xanthan gum, which is a shear-thinning fluid. They find that the contour lengths of the XG molecule decrease with increasing ionic strength of the PBS saline, which has the consequence that XG molecules are clearly affected by the ionic strength of the liquid. The authors perform a particle flow experiments, using PS beads, Comsol simulations, and extensive rheological characterization of the liquids. The paper is well written, and the results are clearly presented in well composed figures. The conclusions seem to be appropriately supported by experiments and modelling. In conclusion, I recommend publication of the ms in its present form.

Thank you for your positive comments. There is no further improvement based on the comments by the reviewer since the reviewer was satisfied with the original version of our manuscript.